# Deep Contextual Video Compression

**Jiahao Li, Bin Li, Yan Lu**
Microsoft Research Asia
{li.jiahao, libin, yanlu}@microsoft.com

## Abstract

Most of the existing neural video compression methods adopt the predictive coding framework, which first generates the predicted frame and then encodes its residue with the current frame. However, as for compression ratio, predictive coding is only a sub-optimal solution as it uses simple subtraction operation to remove the redundancy across frames. In this paper, we propose a deep contextual video compression framework to enable a paradigm shift from predictive coding to conditional coding. In particular, we try to answer the following questions: how to define, use, and learn condition under a deep video compression framework. To tap the potential of conditional coding, we propose using feature domain context as condition. This enables us to leverage the high dimension context to carry rich information to both the encoder and the decoder, which helps reconstruct the high-frequency contents for higher video quality. Our framework is also extensible, in which the condition can be flexibly designed. Experiments show that our method can significantly outperform the previous state-of-the-art (SOTA) deep video compression methods. When compared with x265 using *veryslow* preset, we can achieve 26.0% bitrate saving for 1080P standard test videos. The codes are at `https://github.com/DeepMC-DCVC/DCVC`.

## 1   Introduction

From H.261 [1] developed in 1988 to the just released H.266 [2] in 2020, all traditional video coding standards are based on a predictive coding paradigm, where the predicted frame is first generated by handcrafted modules and then the residue between the current frame and the predicted frame is encoded and decoded. Recently, many deep learning (DL)-based video compression methods [3–11] also adopt the predictive coding framework to encode the residue, where all handcrafted modules are merely replaced by neural networks.

Encoding residue is a simple yet efficient manner for video compression, considering the strong temporal correlations among frames. However, residue coding is not optimal to encode the current frame $x_t$ given the predicted frame $\tilde{x}_t$, because it only uses handcrafted subtraction operation to remove the redundancy across frames. The entropy of residue coding is greater than or equal to that of conditional coding [12]: $H(x_t - \tilde{x}_t) \geq H(x_t|\tilde{x}_t)$, where $H$ represents the Shannon entropy. Theoretically, one pixel in frame $x_t$ correlates to all the pixels in the previous decoded frames and the pixels already been decoded in $x_t$. For traditional video codec, it is impossible to use the handcrafted rules to explicitly explore the correlation by taking all of them into consideration due to the huge space. Thus, residue coding is widely adopted as a special extremely simplified case of conditional coding, with the very strong assumption that the current pixel only has the correlation with the predicted pixel. DL opens the door to automatically explore correlations in a huge space. Considering the success of DL in image compression [13, 14], which just uses autoencoder to explore correlation in image, why not use network to build the conditional coding-based autoencoder to explore correlation in video rather than restricting our vision into residue coding?

35th Conference on Neural Information Processing Systems (NeurIPS 2021).

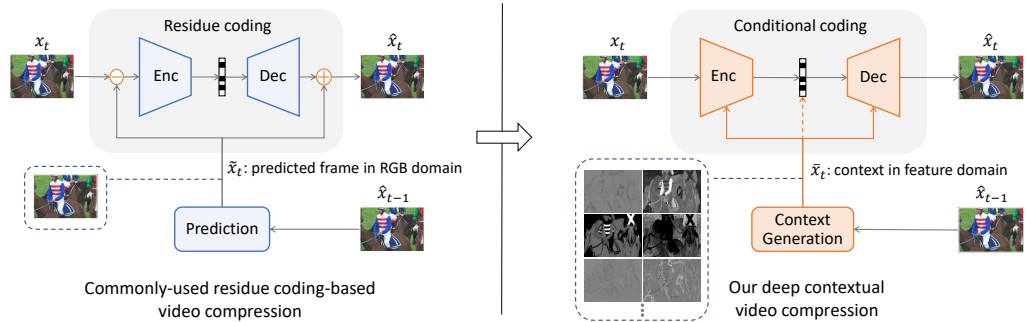

Figure 1: Paradigm shift from residue coding-based framework to conditional coding-based framework. $x_t$ is the current frame. $\hat{x}_t$ and $\hat{x}_{t-1}$ are the current and previous decoded frames. The orange dashed line means that the context is also used for entropy modeling.

When we design the conditional coding-based solution, a series of questions naturally come up: *What is condition? How to use condition? And how to learn condition?* Technically speaking, condition can be anything that may be helpful to compress the current frame. The predicted frame can be used as condition but it is not necessary to restrict it as the only representation of condition. Thus, we define the condition as learnable contextual features with arbitrary dimensions. Along this idea, we propose a deep contextual video compression (DCVC) framework to utilize condition in a unified, simple, yet efficient approach. The diagram of our DCVC framework is shown in Fig. 1. The contextual information is used as part of the input of contextual encoder, contextual decoder, as well as the entropy model. In particular, benefiting from the temporal prior provided by context, the entropy model itself is temporally adaptive, resulting in a richer and more accurate model. As for how to learn condition, we propose using motion estimation and motion compensation (MEMC) at feature domain. The MEMC can guide the model where to extract useful context. Experimental results demonstrate the effectiveness of the proposed DCVC. For 1080p standard test videos, our DCVC can achieve 26.0% bitrate saving over x265 using *veryslow* preset, and 16.4% bitrate saving over previous SOTA DL-based model DVCPro [4].

Actually, the concept of conditional coding has appeared in [15, 16, 12, 17]. However, these works are only designed for partial module (e.g., only entropy model or encoder) or need handcrafted operations to filter which content should be conditionally coded. By contrast, our framework is a more comprehensive solution which considers all of encoding, decoding, and entropy modeling. In addition, the proposed DCVC is an extensible conditional coding-based framework, where the condition can be flexibly designed. Although this paper proposes using feature domain MEMC to generate contextual features and demonstrates its effectiveness, we still think it is an open question worth further investigation for higher compression ratio.

Our main contributions are four-folded:

- We design a deep contextual video compression framework based on conditional coding. The definition, usage, and learning manner of condition are all innovative. Our method can achieve higher compression ratio than previous residue coding-based methods.

- We propose a simple yet efficient approach using context to help the encoding, decoding, as well as the entropy modeling. For entropy modeling, we design a model which utilizes spatial-temporal correlation for higher compression ratio or only utilizes temporal correlation for fast speed.

- We define the condition as the context in feature domain. The context with higher dimensions can provide richer information to help reconstruct the high frequency contents.

- Our framework is extensible. There exists great potential in boosting compression ratio by better defining, using, and learning the condition.

## 2 Related works

**Deep image compression**    Recently there are many works for deep image compression. For example, the compressive autoencoder [18] could get comparable results with JPEG 2000. Subsequently, many works boost the performance by more advanced entropy models and network structures. For example, Ballé *et al.* proposed the factorized [19] and hyper prior [13] entropy models. The method based on hyper prior catches up H.265 intra coding. The entropy model jointly utilizing hyper prior and auto regressive context outperforms H.265 intra coding. The method with Gaussian mixture model [20] is comparable with H.266 intra coding. For the network structure, some RNN (recurrent neural network)-based methods [21–23] were proposed in the early development stage, but most of recent methods are based on CNN (convolutional neural network).

**Deep video compression**    Existing works for deep video compression can be classified into two categories, i.e. non-delay-constrained and delay-constrained. For the first category, there is no restriction on reference frame location, which means that the reference frame can be from future. For example, Wu *et al.* [10] proposed interpolating the predicted frame from previous and future frames, and then the frame residue is encoded. Djelouah *et al.*[8] also followed this coding structure and introduced the optical flow estimation network to get better predicted frame. Yang *et al.* [6] designed a recurrent enhancement module for this coding structure. In addition, 3D autoencoder was proposed to encode group of pictures in [24, 25]. This is a natural extension of deep image compression by increasing the dimension of input. It is noted that this coding manner will bring larger delay and the GPU memory cost will be significantly increased. For the delay-constrained methods, the reference frame is only from the previous frames. For example, Lu *et al.* [3] designed the DVC model, where all modules in traditional hybrid video codec are replaced by networks. Then the improved model DVCPro which adopts more advanced entropy model from [14] and deeper network was proposed in [4]. Following the similar framework with DVC, Agustsson *et al.* designed a more advanced optical flow estimation in scale space. Hu *et al.* [26] considered the rate distortion optimization when encoding motion vector (MV). In [7], the single reference frame is extended to multiple reference frames. Recently, Yang *et al.* [6] proposed an RNN-based MV/residue encoder and decoder. In [11], the residue is adaptively scaled by learned parameter.

Our research belongs to the delay-constrained method as it can be applied in more scenarios, e.g. real time communication. Different from the above works, we design a conditional coding-based framework rather than following the commonly-used residue coding. Other video tasks show that utilizing temporal information as condition is helpful [27, 28]. For video compression, recent works in [15], [16], and [12, 17] have some investigations on condition coding. In [15], the conditional coding is only designed for entropy modeling. However, due to the lack of MEMC, the compression ratio is not high, and the method in [15] cannot outperform DVC in terms of PSNR. By contrast, our conditional coding designed for encoding, decoding, and entropy modeling can significantly outperform DVCPro. In [16], only encoder takes the conditional coding. However, for decoder, the residual coding is still adopted. As a latent state is used, the framework in [16] is difficult to train [7]. By contrast, we use explicit MEMC to guide the context learning, which is easier to train. In [12, 17], the video contents need to be explicitly classified into skip and non-skip modes, where only the contents with non-skip mode use the conditional coding. By contrast, our method does not need the handcrafted operation to decompose the video. In addition, the condition in DCVC is the context in feature domain, which has much larger capacity. In summary, when compared with [15], [16], and [12, 17], the definition, usage, and learning manner of the condition in DCVC are all innovative.

## 3 Proposed method

In this section, we present the details of the proposed DCVC. We first describe the whole framework of DCVC. Then we introduce the entropy model for compressing the latent codes, followed by the approach of learning the context. At last, we provide the details about training.

### 3.1 The framework of DCVC

In traditional video codec, the inter frame coding adopts the residue coding, formulated as:

$$\hat{x}_t = f_{dec}\big(\lfloor f_{enc}(x_t - \tilde{x}_t)\rceil\big) + \tilde{x}_t \; with \; \tilde{x}_t = f_{predict}(\hat{x}_{t-1}). \tag{1}$$

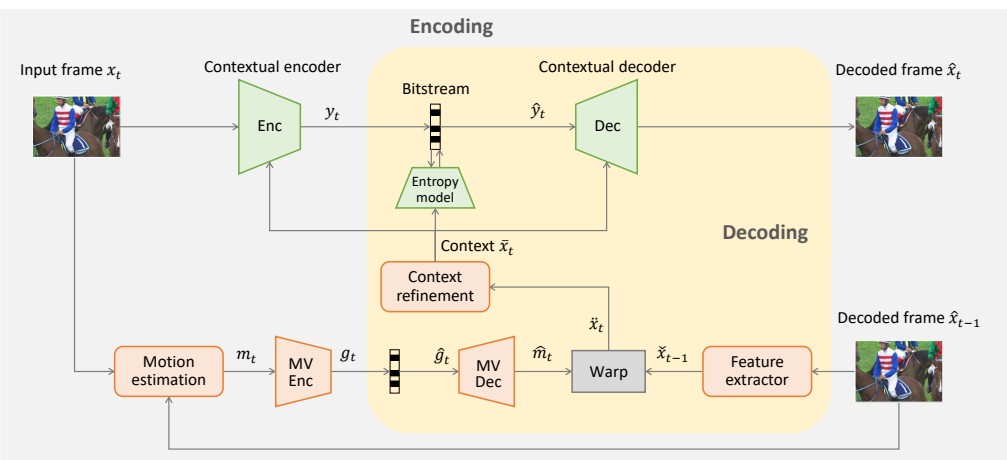

Figure 2: The framework of our DCVC.

$x_t$ is the current frame. $\hat{x}_t$ and $\hat{x}_{t-1}$ are the current and previous decoded frames. For simplification, we use single reference frame in the formulation. $f_{enc}(\cdot)$ and $f_{dec}(\cdot)$ are the residue encoder and decoder. $\lfloor \cdot \rceil$ is the quantization operation. $f_{predict}(\cdot)$ represents the function for generating the predicted frame $\tilde{x}_t$. In traditional video codec, $f_{predict}(\cdot)$ is implemented in the manner of MEMC, which uses handcrafted coding tools to search the best MV and then interpolates the predicted frame. For most existing DL-based video codecs [3–9], $f_{predict}(\cdot)$ is the MEMC which is totally composed of neural networks.

In this paper, we do not adopt the commonly-used residue coding but try to design a conditional coding-based framework for higher compression ratio. Actually, one straightforward conditional coding manner is directly using the predicted frame $\tilde{x}_t$ as the condition:

$$\hat{x}_t = f_{dec}\big(\lfloor f_{enc}(x_t|\tilde{x}_t)\rceil \mid \tilde{x}_t\big) \; with \; \tilde{x}_t = f_{predict}(\hat{x}_{t-1}). \tag{2}$$

However, the condition is still restricted in pixel domain with low channel dimensions. This will limit the model capacity. Now that the conditional coding is used, why not let model learn the condition by itself? Thus, this paper proposes a contextual video compression framework, where we use network to generate context rather than the predicted frame. Our framework can be formulated as:

$$\hat{x}_t = f_{dec}\big(\lfloor f_{enc}(x_t|\bar{x}_t)\rceil \mid \bar{x}_t\big) \; with \; \bar{x}_t = f_{context}(\hat{x}_{t-1}). \tag{3}$$

$f_{context}(\cdot)$ represents the function of generating context $\bar{x}_t$. $f_{enc}(\cdot)$ and $f_{dec}(\cdot)$ are the contextual encoder and decoder, which are different from residue encoder and decoder. Our DCVC framework is illustrated in Fig. 2.

To provide richer and more correlated condition for encoding $x_t$, the context is in the feature domain with higher dimensions. In addition, due to the large capacity of context, different channels therein have the freedom to extract different kinds of information. Here we give an analysis example in Fig. 3. In the figure, the upper right part shows four channel examples in context. Looking into the four channels, we can find different channels have different focuses. For example, the third channel seems to put a lot of emphases on the high frequency contents when compared with the visualization of high frequency in $x_t$. By contrast, the second and fourth channels look like extracting color information, where the second channel focuses on green color and the fourth channel emphasizes the red color. Benefiting from these various contextual features, our DCVC can achieve better reconstruction quality, especially for the complex textures with lots of high frequencies. The bottom right image in Fig. 3 shows the reconstruction error reduction of DCVC when compared with residue coding-based framework. From this comparison, DCVC can achieve non-trivial error decrease on the high frequency regions in both background and foreground, which are hard to compress for many video codecs.

As shown in Fig. 2, the encoding and decoding of the current frame are both conditioned on the context $\bar{x}_t$. Through contextual encoder, $x_t$ is encoded into latent codes $y_t$. $y_t$ is then quantized as $\hat{y}_t$ via rounding operation. Via the contextual decoder, the reconstructed frame $\hat{x}_t$ is finally obtained. In our design, we use network to automatically learn the correlation between $x_t$ and $\bar{x}_t$ and then

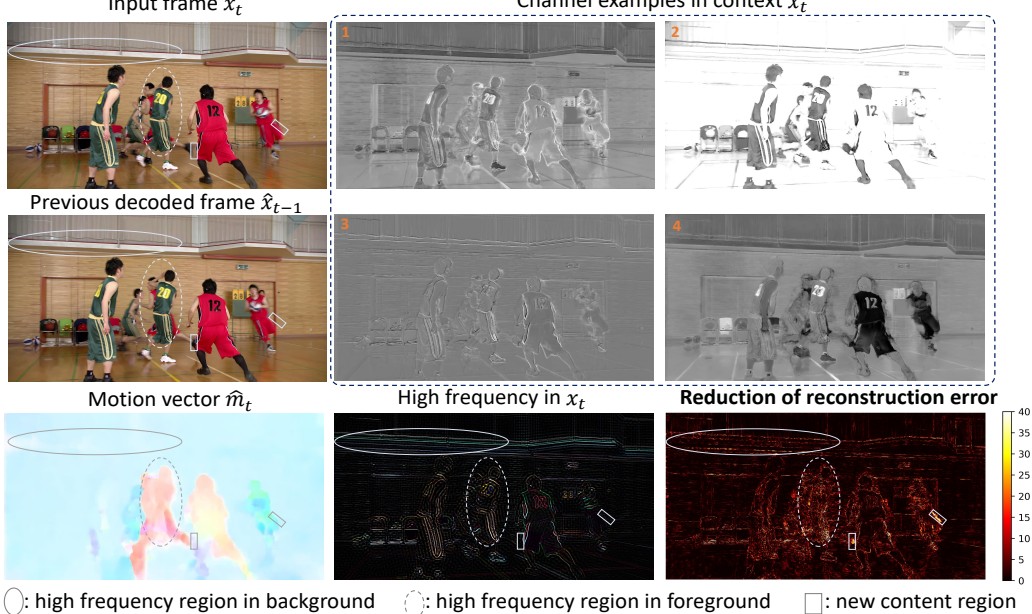

Input frame $x_t$

Channel examples in context $\bar{x}_t$

Previous decoded frame $\hat{x}_{t-1}$

Motion vector $\hat{m}_t$

High frequency in $x_t$

**Reduction of reconstruction error**

⬭: high frequency region in background    ⬭: high frequency region in foreground    ▭: new content region

Figure 3: Visual examples from *videoSRC05* in MCL-JCV[29] dataset. The high frequency in $x_t$ is decomposed by discrete cosine transform. It shows that DCVC improves the reconstruction of high frequency contents in both background with small motion and foreground with large motion. In addition, DCVC is good at encoding the new content region caused by motion, where the reconstruction error can be significantly decreased compared with residue coding-based framework DVCPro[4]. The BPP (bits per pixel) of DCVC (0.0306) is smaller than that of DVCPro (0.0359).

remove the redundancy rather than using fixed subtraction operation in residue coding. From another perspective, our method also has the ability to adaptively use the context. Due to the motion in video, new contents often appear in the object boundary regions. These new contents probably cannot find a good reference in previous decoded frame. In this situation, the DL-based video codec with frame residue coding is still forced to encode the residue. For the new contents, the residue can be very large and the inter coding via subtraction operation may be worse than the intra coding. By contrast, our conditional coding framework has the capacity to adaptively utilize the condition. For the new contents, the model can adaptively tend to learn intra coding. As shown in reconstruction error reduction in Fig. 3, the reconstruction error of new contents can be significantly reduced.

In addition, this paper not only proposes using the context $\bar{x}_t$ to generate the latent codes, but also proposes utilizing it to build the entropy model. More details are introduced in the next subsection.

### 3.2 Entropy model

According to [30], the cross-entropy between the estimated probability distribution and the actual latent code distribution is a tight lower bound of the actual bitrate, namely

$$R(\hat{y}_t) \geq \mathbb{E}_{\hat{y}_t \sim q_{\hat{y}_t}}[-log_2 p_{\hat{y}_t}(\hat{y}_t)], \tag{4}$$

$p_{\hat{y}_t}(\hat{y}_t)$ and $q_{\hat{y}_t}(\hat{y}_t)$ are estimated and true probability mass functions of quantized latent codes $\hat{y}_t$, respectively. Actually, the arithmetic coding almost can encode the latent codes at the bitrate of cross-entropy. The gap between actual bitrate $R(\hat{y}_t)$ and the bitrate of cross-entropy is negligible. So our target is designing an entropy model which can accurately estimate the probability distribution of latent codes $p_{\hat{y}_t}(\hat{y}_t)$. The framework of our entropy model is illustrated in Fig. 4. First, we use the hyper prior model [13] to learn the hierarchical prior and use auto regressive network [14] to learn the spatial prior. The two priors (hierarchical prior and spatial prior) are commonly-used in deep image compression. However, for video, the latent codes also have the temporal correlation. Thus, we propose using the context $\bar{x}_t$ to generate the temporal prior. As shown in Fig. 4, we design a temporal prior encoder to explore the temporal correlation. The prior fusion network will learn to

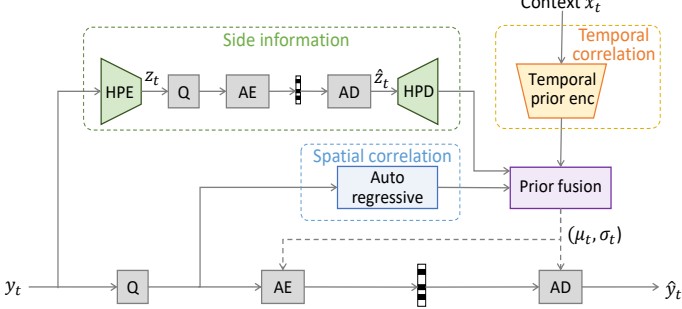

Figure 4: Our entropy model used to encode the quantized latent codes $\hat{y}_t$. HPE and HPD are hyper prior encoder and decoder. Q means quantization. AE and AD are arithmetic encoder and decoder.

fuse the three different priors and then estimate the mean and scale of latent code distribution. In this paper, we follow the existing work [31] and assume that $p_{\hat{y}_t}(\hat{y}_t)$ follows the Laplace distribution. The formulation of $p_{\hat{y}_t}(\hat{y}_t)$ is:

$$p_{\hat{y}_t}(\hat{y}_t|\bar{x}_t, \hat{z}_t) = \prod_i \left( \mathcal{L}(\mu_{t,i}, \sigma_{t,i}^2) * \mathcal{U}(-\frac{1}{2}, \frac{1}{2}) \right)(\hat{y}_{t,i})$$

$$with \ \mu_{t,i}, \sigma_{t,i} = f_{pf}\left( f_{hpd}(\hat{z}_t), f_{ar}(\hat{y}_{t,<i}), f_{tpe}(\bar{x}_t) \right).$$

(5)

The index $i$ represents the spatial location. $f_{hpd}(\cdot)$ is the hyper prior decoder network. $f_{ar}(\cdot)$ is the auto regressive network. $f_{tpe}(\cdot)$ is the specially designed temporal prior encoder network. $f_{pf}(\cdot)$ denotes the prior fusion network. In our entropy model, $f_{ar}(\hat{y}_{t,<i})$ and $f_{tpe}(\bar{x}_t)$ provide the spatial and temporal priors, respectively. $f_{hpd}(\hat{z}_t)$ provides the supplemental side information which cannot be learned from spatial and temporal correlation.

The entropy model formulated in Eq. 5 utilizes spatial prior for higher compression ratio. However, the operations about spatial prior are non-parallel and then lead to slow encoding/decoding speed. By contrast, all operations about the proposed temporal prior are parallel. Thus, we also provide a solution which removes spatial prior but relies on temporal prior for acceleration, namely $\mu_{t,i}, \sigma_{t,i} = f_{pf}\left( f_{hpd}(\hat{z}_t), f_{tpe}(\bar{x}_t) \right)$.

### 3.3 Context learning

For how to learn the context, one alternative solution is directly using a plain network composed by several convolutional layers, where the input is the previous decoded frame $\hat{x}_{t-1}$ and the output is $\bar{x}_t$. However, it is hard for a plain network to extract useful information without supervision. Video often contains various contents and there exist a lot of complex motions. For a position in the current frame, the collocated position in the reference frame may have less correlation. In this situation, the collocated position in context $\bar{x}_t$ is also less correlated to the position in $x_t$, and the less correlated context cannot facilitate the compression of $x_t$. For this reason, we also adopt the idea of MEMC. But different from commonly usage of applying MEMC in pixel domain, we propose performing MEMC in feature domain. The context generation function $f_{context}$ is designed as:

$$f_{context}(\hat{x}_{t-1}) = f_{cr}\left( warp(f_{fe}(\hat{x}_{t-1}), \hat{m}_t) \right)$$

(6)

We first design a feature extraction network $\check{x}_{t-1} = f_{fe}(\hat{x}_{t-1})$ to convert the reference frame from pixel domain to feature domain. At the same time, we use the optical flow estimation network [32] to learn the MV between the reference frame $\hat{x}_{t-1}$ and the current frame $x_t$. The MV $m_t$ is then encoded and decoded. The decoded $\hat{m}_t$ guides the network where to extract the context through warping operation $\ddot{x}_t = warp(\check{x}_{t-1}, \hat{m}_t)$. The $\ddot{x}_t$ is kind of relatively rough context because the warping operation may introduce some spatial discontinuity. Thus, we design a context refinement network $f_{cr}(\cdot)$ to obtain the final context $\bar{x}_t = f_{cr}(\ddot{x}_t)$. In function $f_{context}(\cdot)$, MV $\hat{m}_t$ not only guides the network where to extract the context but also enables network to learn context from a larger reference region when compared with the solution without MEMC.

Table 1: The BD-Bitrate comparison

| Method | MCL-JCV | UVG | HEVC Class B | HEVC Class C | HEVC Class D | HEVC Class E |
|---|---|---|---|---|---|---|
| **DCVC (proposed)** | **-23.9%** | **-25.3%** | **-26.0%** | **-5.8%** | **-17.5%** | **-11.9%** |
| DVCPro [4] | -4.1% | -7.9% | -9.0% | 7.2% | -6.9% | 17.2% |
| x265 (*veryslow*) | 0.0% | 0.0% | 0.0% | 0.0% | 0.0% | 0.0% |
| DVC [3] | 13.3% | 17.2% | 7.9% | 15.1% | 7.2% | 21.1% |
| x264 (*veryslow*) | 32.7% | 30.3% | 35.0% | 19.9% | 15.5% | 50.0% |

The anchor is x265 (*veryslow*). Negative number means bitrate saving and positive number means bitrate increase.

## 3.4 Training

The target of video compression is using least bitrate to get the best reconstruction quality. Thus, the training loss consists of two metrics:

$$L = \lambda \cdot D + R \tag{7}$$

$\lambda$ controls the trade-off between the distortion $D$ and the bitrate cost $R$. $D$ can be MSE (mean squared error) or MS-SSIM (multiscale structural similarity) for different targets. During the training, $R$ is calculated as the cross-entropy between the true and estimated probability of the latent codes.

For the learning rate, it is set as 1e-4 at the start and 1e-5 at the fine-tuning stage. The training batch size is set as 4. For comparing DCVC with other methods, we follow [5] and train 4 models with different $\lambda$ s {MSE: 256, 512, 1024, 2048; MS-SSIM: 8, 16, 32, 64}.

## 4 Experimental results

### 4.1 Experimental settings

**Training data** We use the training part in Vimeo-90k septuplet dataset [33] as our training data. During the training, we will randomly crop videos into 256x256 patches.

**Testing data** The testing data includes HEVC Class B (1080P), C (480P), D (240P), E (720P) from the common test conditions [34] used by codec standard community. In addition, The 1080p videos from MCL-JCV[29] and UVG[35] datasets are also tested.

**Testing settings** The GOP (group of pictures) size is same with [4], namely 10 for HEVC videos and 12 for non-HEVC videos. As this paper only focuses on inter frame coding, for intra frame coding, we directly use existing deep image compression models provided by CompressAI [36]. We use *cheng2020-anchor* [20] for MSE target and use *hyperprior* [13] for MS-SSIM target.

According to the performance comparison in [37, 38, 31], DVCPro[4] is one previous SOTA DL-based codec among recent works [8, 6, 9, 5, 26]. Thus, we compare DVCPro in our paper. In addition, its predecessor DVC[3] is also tested. It is noted that, for fair comparison, DVC and DVCPro are retested using the same intra frame coding with DCVC. For traditional codecs, x264 and x265 encoders [39] are tested. The settings of these two encoders are same with [4] except two options. One is that we use the *veryslow* preset rather than *veryfast* preset. *Veryslow* preset can achieve higher compression ratio than *veryfast* preset. Another is we use the constant quantization parameter setting rather than constant rate factor setting to avoid the influence of rate control.

### 4.2 Performance comparison

**Compression ratio** Fig. 5 and Fig. 6 show the rate-distortion curves among these methods, where the distortion in Fig. 5 is measured by PSNR and the distortion in Fig. 6 is measured by MS-SSIM. From these figures, we can find that our DCVC model can outperform DVCPro for all bitrate ranges. Table 1 gives the corresponding BD-Bitrate [40] results in terms of PSNR. When compared with x265 using *veryslow* preset, DVCPro achieves 4.1%, 7.9%, 9.0%, and 6.9% bitrate saving on MCL-JCV, UVG, HEVC Class B, and D, respectively. However, for HEVC Class C and E, DVCPro performs worse, and there are 7.2% and 17.2% bitrate increases. By contrast, our DCVC can outperform x265 on all

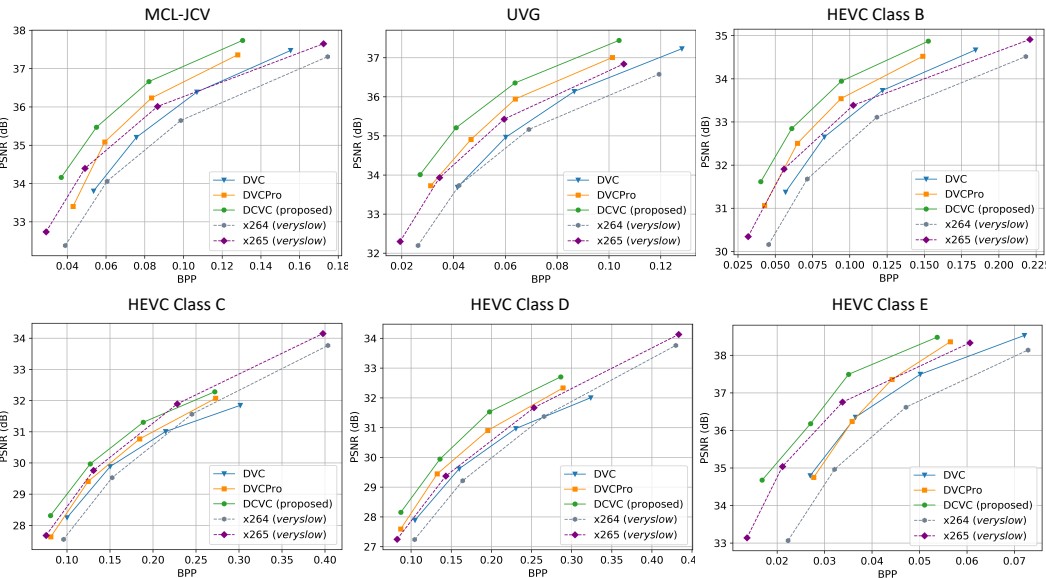

Figure 5: PSNR and bitrate comparison. The horizontal axis is bits per pixel (BPP) representing bitrate cost and vertical axis is PSNR representing reconstruction quality.

datasets. For three 1080p datasets (MCL-JCV, UVG, HEVC Class B), the bitrate savings are 23.9%, 25.3%, and 26.0%, respectively. For low resolution videos HEVC Class C and D, the improvements are also adequate, i.e. 5.8% and 17.5%. For HEVC Class E with relatively small motion, the bitrate saving is 11.9%. From these comparisons, we can find that our DCVC can significantly outperform DVCPro and x265 for various videos with different resolutions and different content characteristics.

In addition, we can find that DCVC can achieve larger improvement on high resolution videos. This is because that high resolution video contains more textures with high frequency. For this kind of video, the feature domain context with higher dimensions is more helpful and able to carry richer contextual information to reconstruct the high frequency contents.

**Complexity** The MACs (multiply–accumulate operations) are 2268G for DCVC and 2014G for DVCPro, and there is about 13% increase. However, the actual inference time per 1080P frame is 857 ms for DCVC and 849 ms for DVCPro on P40 GPU, and there is only about 1% increase, mainly due to the parallel ability of GPU.

### 4.3 Ablation study

**Conditional coding and temporal prior** In our DCVC, we propose using concatenation-based conditional coding to replace subtraction-based residue coding. At the same time, we design the temporal prior for entropy model. To verify the effectiveness of these ideas, we make the ablation study shown in Table 2, where the baseline is our final solution (i.e. temporal prior + concatenating context feature). From this table, we can find that both concatenating RGB prediction and concatenating context feature improve the compression ratio. It verifies the benefit of conditional coding compared with residue coding. In addition, we can find that the improvement of concatenating context feature is much larger than that of concatenating RGB prediction. It shows the advantage of context in feature domain. From Table 2, we also find that the proposed temporal prior further boosts the performance, and its improvement under conditional coding (no matter concatenating RGB prediction or concatenating context feature) is larger than that under residue coding. These results demonstrate the advantage of our ideas.

**Entropy model** In DCVC, besides the hyper prior model, the entropy model for compressing the quantized latent codes $\hat{y}_t$ utilizes both spatial and temporal priors for higher compression ratio. However, the drawback of spatial prior is slow encoding/decoding speed as it brings spatial dependency and is non-parallel. By contrast, all operations about the proposed temporal prior are parallel. Thus, our DCVC also supports removing spatial prior but relies on temporal prior for acceleration.

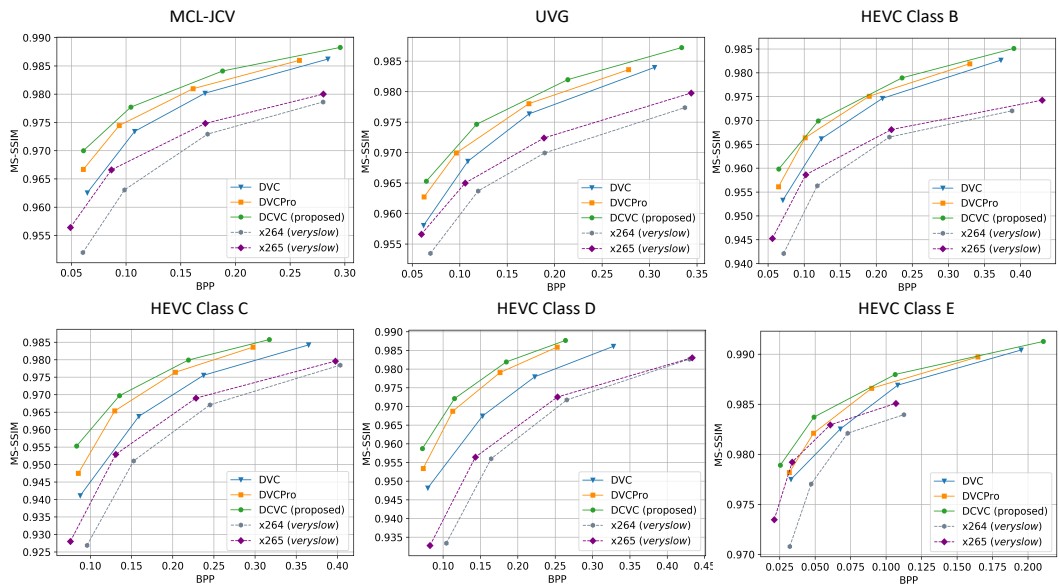

Figure 6: MS-SSIM and bitrate comparison. The DL-based codecs are fine-tuned for MS-SSIM.

Table 2: Ablation study on conditional coding and temporal prior

| Temporal prior | Concatenate context feature | Concatenate RGB prediction | Bitrate increase |
|:---:|:---:|:---:|:---:|
| ✓ | ✓ | | 0.0% |
| ✓ | | ✓ | 5.4% |
| | ✓ | | 4.6% |
| | | ✓ | 8.7% |
| ✓ | | | 11.2% |
| | | | 12.9% |

Benefiting from the rich temporal context, the model without spatial prior only has small bitrate increase. Table 3 compares the performance influence of spatial and temporal priors. From this table, we can find that the performance has large drop if both priors are disabled. When enabling either of these two priors, the performance can be improved a lot. When enabling both of them, the performance can be further improved. However, considering the trade-off between complexity and compression ratio, the solution only using hyper prior and temporal prior is better. These results show the advantage of our temporal prior-based entropy model.

## 4.4 Influence of intra frame coding and comparison with more baselines

To build the best DL-based video compression framework, we use the SOTA DL-based image compression as our intra frame coding. It is noted that, for fair comparison, DVC and DVCPro are retested using the same intra frame coding with DCVC, and their results are better than that reported in

Table 3: Ablation study on entropy model

| Entropy model | Bitrate increase |
|:---|:---:|
| hyper prior + spatial prior + temporal prior | 0.0% |
| hyper prior + temporal prior | 3.8% |
| hyper prior + spatial prior | 4.6% |
| hyper prior | 60.9% |

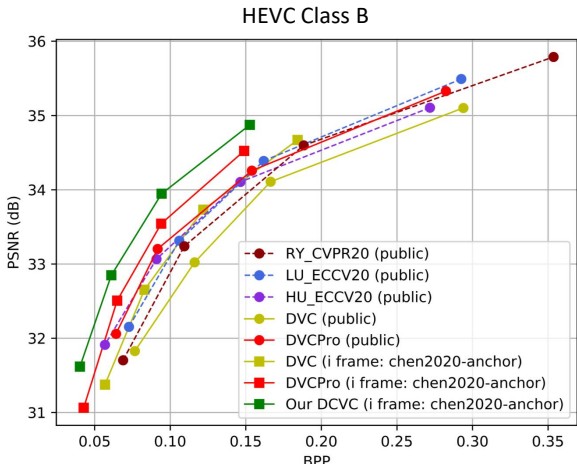

Figure 7: Performance comparison with the public results. The results of method with suffix *(public) are provided by [38, 31]. The results of method with suffix *(i frame: cheng2020-anchor) use *cheng2020-anchor* as the intra frame coding.

their original papers. Fig. 7 shows the performance comparison between our retested DVC/DVCPro and their public results provided by [38, 31]. In addition, Fig. 7 also shows the results of the recent works RY_CVPR20 [6], LU_ECCV20 [5], and HU_ECCV20[26], provided by [38, 31]. From the public results in this figure, we can find that DVCPro is one SOTA method among recent works. The results of LU_ECCV20 and HU_ECCV20 are quite close with DVCPro. When the intra frame coding of DVC and DVCPro uses SOTA DL-based image compression model *cheng2020-anchor* provided by CompressAI [36], their performance has large improvement. When using the same intra frame coding, our proposed DCVC method can significantly outperform DVCPro, as shown in Fig. 7.

## 5 Discussion

In this paper, we make efforts on designing a conditional coding-based deep video compression framework which has a lower entropy bound than the commonly-used residue coding-based framework. Residue coding-based framework assumes the inter frame prediction is always most efficient, which is inadequate, especially for encoding new contents. By contrast, our conditional coding enables the adaptability between learning temporal correlation and learning spatial correlation. In addition, the condition is defined as feature domain context in DCVC. Context with higher dimensions can provide richer information to help the conditional coding, especially for the high frequency contents. In the future, high resolution video is more popular. High resolution video contains more high frequency contents, which means the advantage of our DCVC will be more obvious.

When designing a conditional coding-based framework, the core questions are *What is condition? How to use condition? And how to learn condition?* In this paper, DCVC is a solution which answers these questions and demonstrates its effectiveness. However, these core questions are still open. Our DCVC framework is extensible and worthy more investigation. There exists great potential in designing a more efficient solution by better defining, using and learning the condition.

In this paper, we do not add supervision on the channels in context during the training. There may exist redundancy across channels, and this is not conducive to making full advantage of context with high dimensions. In the future, we will investigate how to eliminate the redundancy across the channels to maximize the utilization of context. For context generation, this paper only uses single reference frame. Traditional codecs have shown that using more reference frames can significantly improve the performance. Thus, how the design the conditional coding-based framework given multiple reference frames is very promising. In addition, we currently do not consider temporal stability of reconstruction quality, which can be further improved by post processing or additional training supervision (e.g., loss about temporal stability).

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
