# Supplementary Material to Deep Contextual Video Compression

**Jiahao Li, Bin Li, Yan Lu**
Microsoft Research Asia
{li.jiahao, libin, yanlu}@microsoft.com

This document provides the supplementary material to our proposed deep contextual video compression (DCVC), including detailed network structures, training strategies, as well as additional experimental results to demonstrate the effectiveness of the proposed DCVC.

## 1 Network Architecture

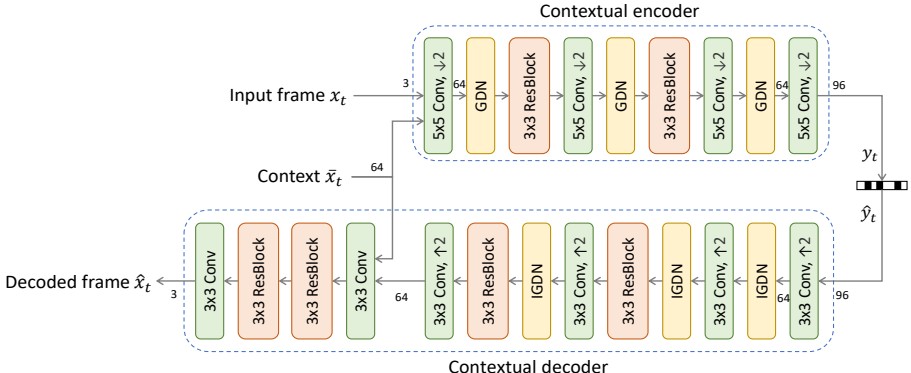

Figure 1: Network structure of contextual encoder and decoder. The above part is the encoder and the below part is the decoder. For simplification, the entropy model is omitted. GDN is generalized divisive normalization [1] and IGDN is the inverse GDN. ResBlock represents plain residual block. The numbers represent channel dimensions.

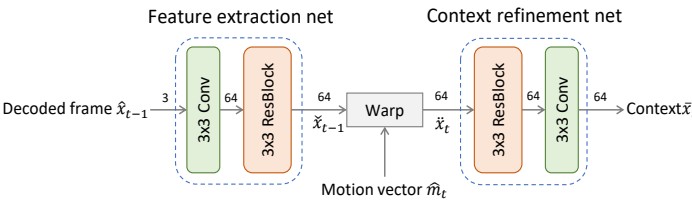

Figure 2: Network structure of feature extraction network and context refinement network.

**Contextual encoder and decoder** The structures of our contextual encoder and decoder are illustrated in Fig. 1. For the contextual encoder, the input is the concatenation of the current frame $x_t$ and context $\bar{x}_t$. The contextual encoder encodes the concatenated data into 16x down-sampled latent codes with dimension 96. For the contextual decoder, we first up-sample the latent codes into the feature with original resolution. Then the up-sampled feature concatenated with context $\bar{x}_t$ is used to generate the final reconstruction frame $\hat{x}_t$.

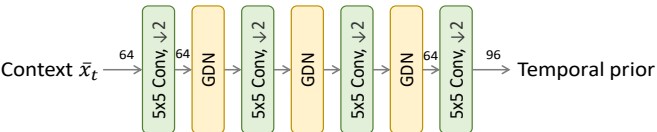

Figure 3: Network structure of temporal prior encoder network, same with the commonly used encoder in image compression [2] except the channel dimensions.

Table 1: The training loss used in progressive training

| Step | Loss | Calculation |
|------|------|-------------|
| 1 | $L_{me}$ | $\lambda \cdot D(x_t, \tilde{x}_t) + R(\hat{g}_t) + R(\hat{s}_t)$ |
| 2 | $L_{reconstruction}$ | $\lambda \cdot D(x_t, \hat{x}_t)$ |
| 3 | $L_{contextual\_coding}$ | $\lambda \cdot D(x_t, \hat{x}_t) + R(\hat{y}_t) + R(\hat{z}_t)$ |
| 4 | $L_{all}$ | $\lambda \cdot D(x_t, \hat{x}_t) + R(\hat{y}_t) + R(\hat{z}_t) + R(\hat{g}_t) + R(\hat{s}_t)$ |

**Feature extraction and context refinement** Fig. 2 shows the structures of our feature extraction network and context refinement network. Both of them contains a convolution layer and a residual block. Consider the complexity, we do not use deeper network at present.

**Motion vector generation** The motion vector (MV) generation part contains the motion estimation, MV encoder, and decoder. For motion estimation, we use optical flow estimation network [3] to generate MV, like DVCPro [4]. The network structures of MV encoder and decoder (decoder also contains a MV refine network) are same with those in DVCPro [4].

**Entropy model** In the entropy model for compressing the quantized latent codes $\hat{y}_t$, the temporal prior encoder network is borrowed from the encoder in image compression [2] and consists of plain convolution layers (stride is set as 2 for down-sampling) and GDN [1], as shown in Fig. 3. The hyper prior encoder/decoder, auto regressive network, and prior fusion network follow the entropy model in image compression [2]. In addition, the MV latent codes also have corresponding entropy model, where we only use auto regressive model and hyper prior model. There is no temporal prior encoder in the the entropy model for MV latent codes.

## 2 Progressive training

The training loss consists of two metrics, i.e. the distortion $D$ and the bitrate cost $R$. In our method, the bitstream contains four parts, namely $\hat{y}_t$, $\hat{g}_t$, $\hat{z}_t$, and $\hat{s}_t$. $\hat{y}_t$ and $\hat{g}_t$ are the quantized latent codes of the current frame and MV, respectively. $\hat{z}_t$ and $\hat{s}_t$ are their corresponding hyper priors. Thus, the total rate-distortion loss $L_{all}$ should contain the bitrate costs of these four parts. The calculation manner of $L_{all}$ is shown in Table 1.

While we already use the pre-trained optical flow estimation network [3] as the initialization of motion estimation, the training may be still unstable if we directly use $L_{all}$ at the initial stage. Sometimes, the MV bitrate cost is very small but the total rate-distortion loss is large. This is because that the model thinks directly learning to generate context without MEMC is easier. Inspired by the existing work [5] where the progressive training strategy is used, we customize a progressive training strategy for our framework. The training is divided into four steps and the training loss for each step is shown in Table 1:

**Step 1.** Warm up the MV generation part including motion estimation, MV encoder and decoder. The training loss is $L_{me}$. In Table 1, $\tilde{x}_t$ is the warped frame in pixel domain, namely using $\hat{m}_t$ to do warping operation on $\hat{x}_{t-1}$.

**Step 2.** Train other modules except the MV generation part. At this step, the parameters of MV generation part are frozen. The training loss is $L_{reconstruction}$. It means that we only pursue high reconstruction quality. This step is helpful for model to generate context which can better reconstruct the high frequency contents.

**Step 3.** Based on previous step, the bit cost is considered, and the training loss becomes $L_{contextual\_coding}$. This step can be regarded as whole framework training with only freezing the MV generation part.

**Step 4.** Reopen the MV generation part and perform the end-to-end training of whole framework according to $L_{all}$.

The proposed progressive training strategy can stabilize the model training. For the training time, currently we need about one week on single Tesla V100 GPU. We will develop more advanced training technology to shorten the training time in the future. In addition, it is noted that we also apply the progressive strategy for DVCPro (it does not have released models and we retrain it). For DVC, we just use the released models [6].

## 3 Details of experimental settings

**Dataset** The training dataset comes from Vimeo-90k septuplet dataset [7] (MIT License[1]). The testing data includes MCL-JCV dataset [8] (copyright can be found from this link [2]), UVG dataset[9] (BY-NC license[3]), and HEVC standard test videos (more details can be found in [10]). These datasets are commonly-used for video compression research and can be downloaded from Internet. The consents of these datasets are public. In addition, we have manually checked that these datasets do not contain personally identifiable information or offensive content.

**Intra frame coding** The intra frame coding in our framework directly uses the existing deep image compression models, where the model parameters are provided by CompressAI [11] (Apache License 2.0[4]). We use *cheng2020-anchor* [12] for MSE target and use *hyperprior* [13] for MS-SSIM target, as they are the best models provided by CompressAI.

In DCVC, we train 4 models with different $\lambda$ s {MSE: 256, 512, 1024, 2048; MS-SSIM: 8, 16, 32, 64}. The models with quality index 3, 4, 5, 6 (trained with 4 different $\lambda$ s) in CompressAI are used for the corresponding intra frame coding. For example, the model with quality index 6 in CompressAI is used for our DCVC model with $\lambda$ 2048 (for MSE target) or 64 (for MS-SSIM target).

**FFMPEG settings** We test the x264 and x265 encoders from FFMPEG[14]. The settings of these two encoders are same with [4] except two options. One is that we use the *veryslow* preset rather than *veryfast* preset. *Veryslow* preset can achieve higher compression ratio than *veryfast* preset. Another is that we use the constant quantization parameter setting rather than constant rate factor setting, where constant quantization parameter setting can avoid the influence of rate control. The detailed configurations of x264 and x265 are

- x264: *ffmpeg -pix fmt yuv420p -s WxH -r FR -i Video.yuv -vframes N -c:v libx264 -preset veryslow -tune zerolatency -qp QP -g GOP -bf 2 -b strategy 0 -sc threshold 0 output.mkv*
- x265: *ffmpeg -pix fmt yuv420p -s WxH -r FR -i Video.yuv -vframes N -c:v libx265 -preset veryslow -tune zerolatency -x265-params "qp=QP:keyint=GOP" output.mkv*

*W, H, FR, N, QP* and *GOP* represent the width, height, frame rate, the number of encoded frames, quantization parameter, and group of pictures, respectively.

**GOP size and tested frame number** We follow [4] and set the GOP size as 10 for HEVC test videos and 12 for non-HEVC test videos, respectively. The tested frame number of HEVC videos is 100 (10 GOPs), same with [4]. As there is no description about the tested frame number for non-HEVC test videos in [4], we test 120 frames for MCL-JCV and UVG datasets, which has 10 GOPs, same with HEVC test videos.

## 4 Test on larger GOP size

In the paper, we follow [4] and set the GOP size as 10 for HEVC test videos and 12 for non-HEVC test video, denoted as default GOP setting. Actually, this GOP setting is relatively small when compared

---

[1]https://github.com/anchen1011/toflow/blob/master/LICENSE
[2]http://mcl.usc.edu/mcl-jcv-dataset/
[3]https://creativecommons.org/licenses/by-nc/3.0/deed.en_US
[4]https://github.com/InterDigitalInc/CompressAI/blob/master/LICENSE

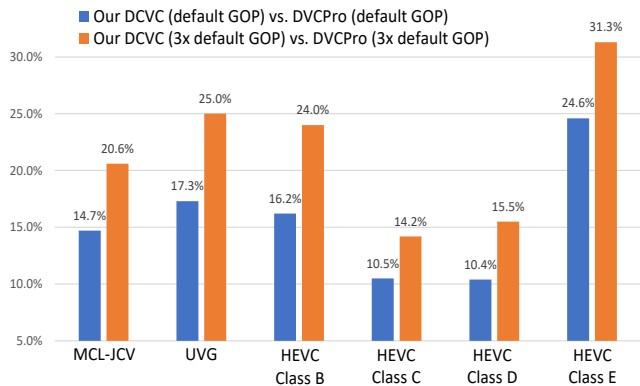

Figure 4: Bitrate saving under different GOP settings. Default GOP setting is {HEVC test videos: 10, non-HEVC test videos: 12}, same with [4]. 3x default GOP setting is {HEVC test videos: 30, non-HEVC test videos: 36}. The tested frame number under two GOP settings is {HEVC test videos: 30, non-HEVC test videos: 36}. We can find that the bitrate saving of our DCVC is larger under larger GOP size.

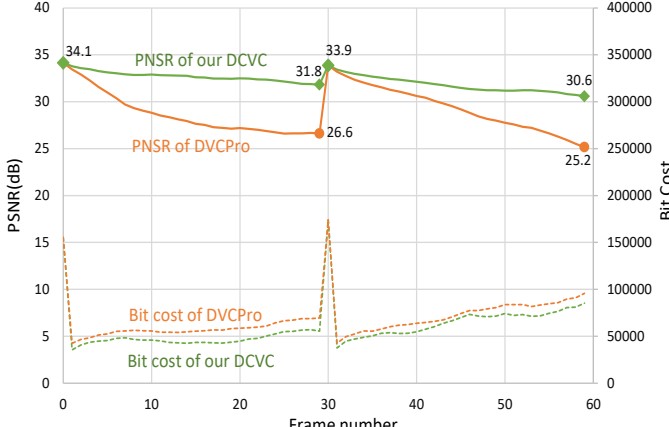

Figure 5: Example of PSNR and bit cost comparison between our DCVC and DVCPro. The tested video is *BasketballDrive* from HEVC Class B dataset. The GOP size is 30, and tested frame number is 60. From this example, we can find that our DCVC can efficiently alleviate the error-propagation problem. DCVC can use fewer bits while achieving much better reconstruction quality.

that in practical scenarios. For this reason, we conduct the experiments under larger GOP size. The bitrate saving comparison is shown in Fig. 4. In this comparison, we increase the GOP size to 3 times of default GOP setting, i.e. 30 for HEVC test videos and 36 for non-HEVC test videos. From this comparison, we can find that, when compared with DVCPro, the improvement of our DCVC is much larger under 3x default GOP size. It shows that our conditional coding-based framework can better deal with the error-propagation problem. Under large GOP size, residue coding still assumes that the inter frame prediction is always most efficient even when the quality of reference frame is bad, then suffers from the large prediction error. By contrast, our conditional coding does not need to pursue the strict equality between prediction frame and the current frame, and enables the adaptability between learning temporal correlation and learning spatial correlation. Thus, the advantage of our DCVC will be more obvious when the GOP size increases. In addition, the bitrate saving increase is larger for high resolution videos. For example, for the 240P dataset HEVC Class D, the bitrate saving is changed from 10.4% to 15.5%. By contrast, for the 1080P dataset HEVC Class B, the bitrate saving is changed from 16.2% to 24.0%. It is because that the context in feature domain is helpful for reconstructing the high frequency contents, then the reconstruction quality can be improved and it is conducive to alleviating error-propagation problem.

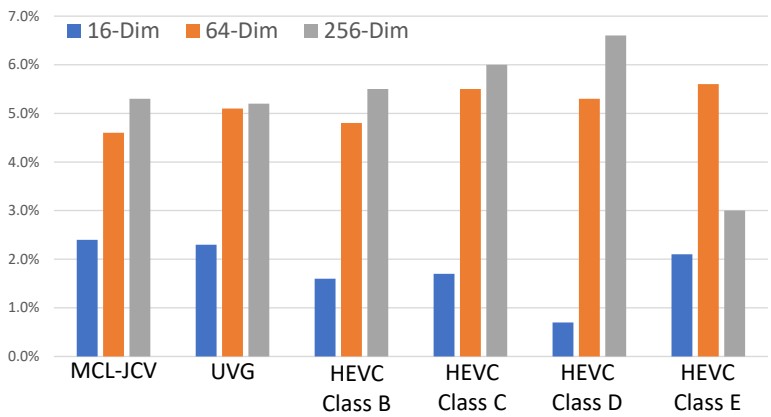

Figure 6: Bitrate saving when using different channel dimensions for context. The anchor is 3-Dim (dimension is 3) model.

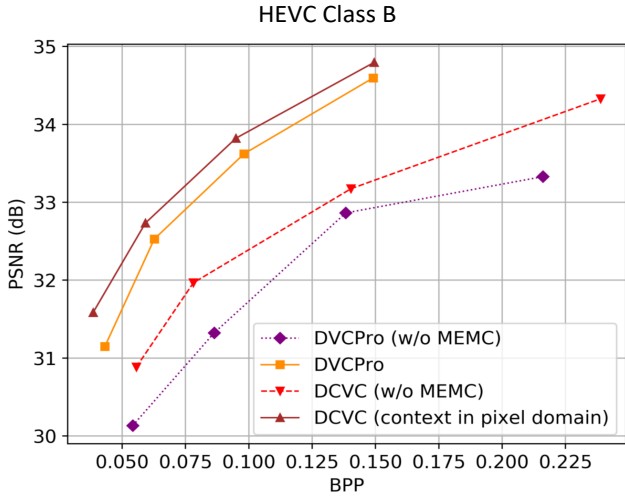

Figure 7: Performance comparison when MEMC is disabled. DCVC (context in pixel domain) refers the model using temporal prior and concatenating RGB prediction.

An example of PSNR and bit cost comparison between our DCVC and DVCPro is shown in Fig. 5. In the example, the PSNR of DVCPro decreases from 34.1 dB to 26.6 dB in the first GOP. By contrast, our DCVC only decreases to 31.8 dB.

## 5 Ablation study

**Channel dimension of context** In DCVC, the channel dimension of context is set as 64 in the implementation. We also conduct the experiment when different dimensions are used. The cases (3, 16, 256-Dim) are also tested. The corresponding bitrate saving comparison is shown in Fig. 6, where the anchor is the 3-Dim model. From this figure, we can observe that the 16-Dim model can improve the performance in some degree, and the 64-Dim model can further boost the performance in a larger degree for most datasets. However, the improvement brought by 256-Dim model is relatively small. The HEVC Class E even has performance loss. The reason may be that the model training is not stable if there is no extra supervision for training the context with so high dimensions and in original resolution. For this reason, we adopt the 64-Dim model at present.

**Motion estimation and motion compensation (MEMC)** In our DCVC, we use MEMC to guide the model where to extract context. Actually we are also very interested in the case without MEMC.

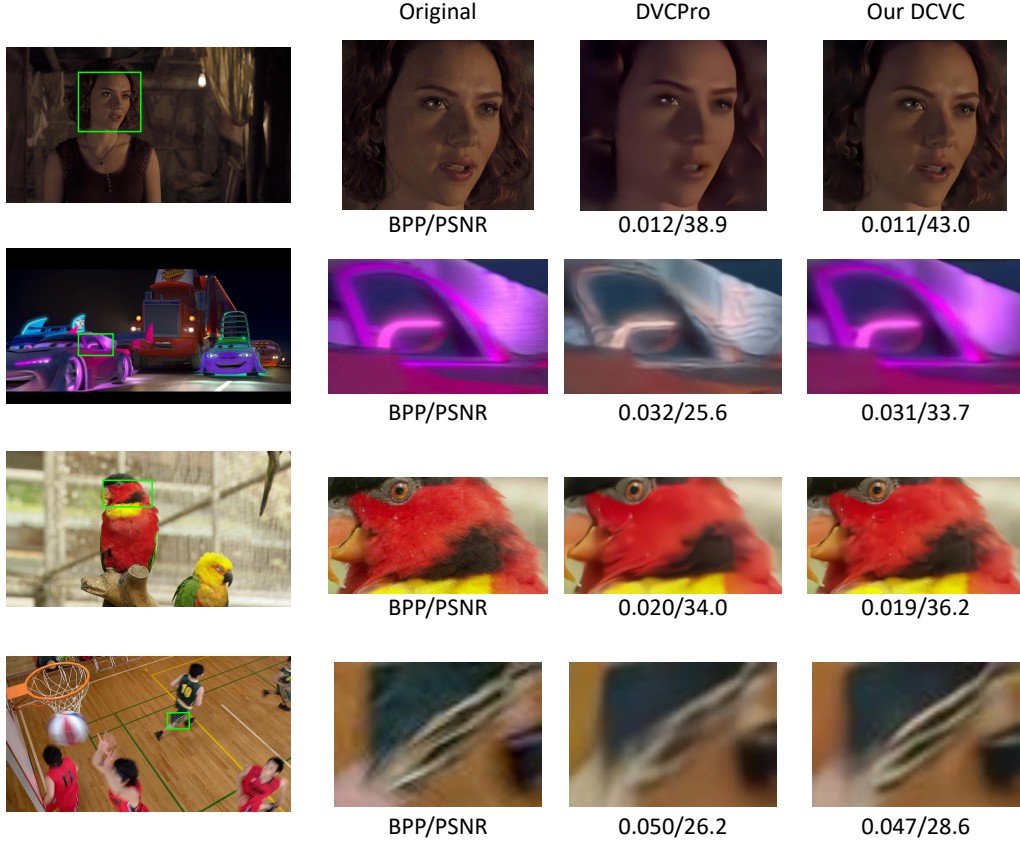

| | Original | DVCPro | Our DCVC |
|---|---|---|---|
| BPP/PSNR | | 0.012/38.9 | 0.011/43.0 |
| BPP/PSNR | | 0.032/25.6 | 0.031/33.7 |
| BPP/PSNR | | 0.020/34.0 | 0.019/36.2 |
| BPP/PSNR | | 0.050/26.2 | 0.047/28.6 |

Figure 8: Examples of visual comparison. The first column shows the original full frames. The second column shows the cropped patch in original frame. The contents in third and fourth columns are reconstructed by DVCPro and our DCVC, respectively.

For these reason, we test the DCVC and DVCPro where the MEMC is removed (directly use the previous decoded frame as the predicted frame in DVCPro and the condition in DCVC). They are denoted as DVCPro (w/o MEMC) and DCVC (w/o MEMC), respectively. As DCVC (w/o MEMC) uses the previous decoded frame as condition, we use the model DCVC (context in pixel domain, i.e. temporal prior + concatenating RGB prediction) for fair comparison. Fig. 7 shows the results. From this figure, we can find that the performance has a large drop for both of DVCPro and DCVC if MEMC is removed. However, DCVC (w/o MEMC) is still better than DVCPro (w/o MEMC), and the performance gap is even larger. The DCVC (context in pixel domain) has 12.7% improvement over DVCPro. By contrast, DCVC (w/o MEMC) can achieve 22.1% bitrate saving compared with DVCPro (w/o MEMC). These results show that the MEMC is helpful for both frame residue coding and conditional coding-based frameworks. When MEMC is disable, the improvement of conditional coding can be larger.

While we currently use MEMC to learn the context, there still exists great potential in designing a better learning manner. In the future, we will continue the investigation. For example, transformer [15] can be used to explore the global correlations and generate the context with larger receptive field.

## 6 Visual comparison

We also conduct the visual comparison between the previous SOTA DVCPro and our DCVC. Several examples are shown in Fig. 8. From these examples, we can find that our DCVC can achieve much higher reconstruction quality without increasing the bitrate cost. For instance, in the example shown in the second row in Fig. 8, we can find that the image reconstructed by DVCPro has obvious color distortion and unexpected textures. By contrast, our DCVC can achieve much better results. In

the example shown in the fourth row, our DCVC also produces much clearer stripe texture in the basketball clothes.