# OpenReview forum: "Deep Contextual Video Compression"
_NeurIPS.cc/2021/Conference — NeurIPS 2021 Poster_

### Official Review · Reviewer_E81t · 2021-07-06

**Rating:** 8
**Confidence:** 4

**Summary:**

The paper proposes to replace the minus/plus operations commonly used in neural video compression (to calculate/add the residual) with neural networks. The resulting contextual encoder/decoder networks are conditioned on the same context that is used for the entropy model, which is calculated by warping features extracted from the previously decoded frame.

**Limitations And Societal Impact:**

The authors have not addressed societal impact.

**Main Review:**

[Post rebuttal update: all my concerns were addressed].

The paper contains a solid idea (replacing minus/plus) and shows promising R-D results. My issues with the paper are:

a) Ablation studies: from the text it is not clear how the proposed method relates to "DVCPro", and Section 4.3. is written unclearly. Is the only difference that the proposed method uses contextual encoder/decoder nets and the generation net? If that is the case, the presented ablation studies make sense, but this has to be made more clear. I.e. I need a paragraph on what exactly the differences between the "DVCPro" and the "DCVC (context in pixel domain)" lines in Fig. 7 are. I am hoping that the second line can be renamed to sth like "DVCPro + context in pixel domain". If there are more novel components, the ablation study is not useful. In particular, I'm wondering what is meant with "DVC and DVCPro are retested using the same intra frame coding" (L229) and whether the training strategy mentioned in Section 2 of the Appendix is used for all?

b) Runtime concerns: there is only one line on complexity (L252). It is unclear which complexity is meant (number of weights, FLOPS, fps, ...?). I would expect the feature extraction and refinement operations (which happen at full res) to be costly, and they are needed for encoding and decoding. As such, I'm surprised how the "complexity" can be the same as DVCPro?

c) Contributions list is overly vague and it's unclear which things are novel. "[...] jointly utilizing the spatial and temporal correlation, [...]" (L66) does not seem novel. "[...] more accurate probability estimation can be obtained" (L67) is not ablated in the text. "We define the condition as the context in feature domain" seems vaguely familiar to what Rippel et al. were doing in their 2018 paper.

d) Entropy Coding? L169: "the gap between the actual bitrate ... and the cross-entropy is negligible": not always, depends on the precision of your arithmetic. This statement makes me wonder whether the presented graphs are using the real, entropy coded bitrate?

e) ffmpeg settings: I did not follow why the arguments "-r", and "-vframes" are needed.

f) Various issues with the text:
- L8 and throughout: I think what is called "condition" should be called "conditioning signal".
- L28: Reference [11] does not contain the entropy formula, and intuitively it is unclear how the entropy of the substraction is higher, I think entropy is the wrong concept to be used here, or I'm missing something in which case clarification is needed.
- L122: "MEMC which is totally composed of network": not sure what this means or how it's different from the proposed method.
- L167, L209: the "true" probability mass function of y is of course unknown, we are using the empirical distribution to estimate the rate. Similarily, the statement on L169, "the gap betweeen the actual bitrate ... and the cross-entropy ..."
- L179: Any reason/intuition for using Laplace?
- L186ff seem similar to L133ff? What is meant with a "plain" network on L186?

Summary:
Overall I like the idea, but I think the paper does not sufficiently place it within the context of previous works.

**Time Spent Reviewing:**

5

---

> ### Author Response · Authors · 2021-08-10
> **Response to the comments from reviewer E81t**
>
> Many thanks for your valuable comments and the recognition on our results. We hope the following responses could address your concerns.
>
> 1. **Unclear ablation studies.**
>
>     Thanks for your suggestion, and we will reorganize Section 4.3. In the original draft, the differences between DCVC (context in pixel domain) and DVCPro include temporal prior and concatenating RGB prediction. Considering another reviewer's question about the influence of temporal prior and concatenation operation, we make more detailed ablation studies, as shown below. In the revised paper, we will use more accurate description for the ablation study.
>
>     |Temporal prior|Concatenate context feature|Concatenate RGB prediction |Bitrate increase|
>     |    :-:       |           :-:             |           :-:             |       :-:      |
>     |     Y        |            Y              |                           |       0.0%     |
>     |     Y        |                           |            Y              |       5.4%     |
>     |              |            Y              |                           |       4.6%     |
>     |              |                           |            Y              |       8.7%     |
>     |     Y        |                           |                           |       11.2%    |
>     |              |                           |                           |       12.9%    |
>     |              |                           |                           |                |
>
> 2. **What is meant with "DVC and DVCPro are retested using the same intra frame coding"**
>
>    The main difference between DVC/DVCPro and our DCVC lies in the inter frame coding. For fair comparison, we tested DVC/DVCPro and our DCVC using the same intra frame coding method.  If compared with the results of DVC/DVCPro in their papers, our method can achieve larger bitrate saving. Actually we tried different intra frame coding methods (e.g., other DL-based image coding and H.265 intra coding), and we found that our DCVC's relative bitrate saving over DVC/DVCPro is similar.
>
> 3. **Training strategy for different neural networks.**
>
>     We apply the same training strategy for both DVCPro (it does not have released models) and our DCVC. For DVC, we just use the released models. We will clarify it in the revised paper.
>
> 4. **It is unclear which complexity is meant.**
>
>     Thanks for pointing it out. In the original paper, the complexity is measured by the model's inference time on P40 GPU. The inference time per 1080P frame is 857 ms for DCVC and 849 ms for DVCPro, and there is about 1% increase. We agree the theoretical analysis on MACs is more accurate in terms of complexity measurement. The MACs are 2268G for DCVC and 2014G for DVCPro, and there is about 13% increase. Although the increase of MACs is 13%, the impact of inference time is only about 1%, mainly due to the parallel ability of GPU. We will add it in the revised paper.
>
> 5. **Contributions list is overly vague.**
>
>     We will revise the paper to clarify the main contributions. We hope the following responses could address your questions.
>
>     - *Spatial-temporal adaptive entropy model*:  It is important for video compression. We design an entropy model which utilizes spatial-temporal correlation for higher compression ratio or only utilizes temporal correlation for fast speed.
>
>     - *"more accurate probability estimation can be obtained" is not ablated in the text*: The ablation study is in the supplementary material (Figure 8). We will move this part to the main paper in the revised paper.
>
>     - *Main differences with Rippel et al.’ work*:
>         - In Rippel et al.’ work, only encoder takes the conditional coding. However, for decoder, the residual coding is still adopted. By contrast, our decoder is also based on conditional coding.
>         - Rippel et al. used a latent state for encoder's conditional coding. As pointed in Lin et al.’ paper [7], Rippel et al.’ framework involving latent state is difficult to train. By contrast, we use explicit MEMC to guide the context learning, which is easier to train.
>
>         We will add this comparison in the revised paper.
>
> 6. **Are the presented graphs using the real or entropy coded bitrate?**
>
>     We did write the streams to file and read the stream from file for decoding. We use arithmetic coder with 16-bit precision and find that the bit calculated from entropy and the actual bitstream size are quite close (the gap is only about 1%).
>
> 7. **ffmpeg settings.**
>
>     We follow [4] to set ffmpeg. ''-r'' controls the frame rate. As we calculate bpp, ''-r'' does not have impact on the result. ''-vframes'' controls the tested frame number.
>
> 8. **Question on entropy formula in L28.**
>
>     The equation $H\left(x_t\middle|{\widetilde{x}}_t\right)\le H(x_t-{\widetilde{x}}_t)$ could be found in Equation (4) in reference [11]. It is based on the theorem that conditional entropy is no larger than independent entropy: $H\left(x_t\middle|{\widetilde{x}}_t\right)=H\left(x_t-{\widetilde{x}}_t\middle|{\widetilde{x}}_t\right)\le H(x_t- {\widetilde{x}}_t)$.
>
> 9. **Laplace distribution in entropy model.**
>
>     There is no special reason for using Laplace distribution. The released code in DVC [27] uses Laplace distribution and we followed it. We also tested Gaussian distribution, and the difference is marginal.
>
> 10. **Unclear statements.**
>
>     - *"MEMC which is totally composed of network"*: It is used to describe the difference between traditional codec and DL-based codec. In traditional codec, MEMC is implemented by hand-crafted rules. In DL-based codec, MEMC composes of neural networks.
>
>     - *"plain" network*: It refers to directly using simple network with several convolutional layers to extract features from previous frame without considering the motion.
>
>     Thanks for your careful review, and we will also make other modifications as you suggested.

---

> > ### Comment · Reviewer_E81t · 2021-08-18
> > **Reply**
> >
> > Thanks for the rebuttal, all my concerns were addressed! I like the ablation study presented. Adding all the information to the paper will make it fairly solid and worthy of publication. I updated my rating.

---

### Official Review · Reviewer_JwvE · 2021-07-13

**Rating:** 7
**Confidence:** 4

**Summary:**

In this work the authors propose a new model for video compression that, instead of warping the previous frame and predicting the current frame, extract and warp features using the motion vectors. As a result, the encoder doesn’t rely on residual compression (residual between ground truth and prediction) but directly encodes the necessary information needed to decode the frame.

**Limitations And Societal Impact:**

The authors discussed some limitations, in particular the fact that the proposed solution is another conditioning which is slightly more complex and rich than image prediction but other alternatives remain to be explored. I think the discussion could be extended to other considerations like temporal stability.

**Main Review:**

**Originality**

Such an approach to video compression is relatively new as most related works indeed focus on predicting a full frame then encoding a residual image. It has however strong links with frame interpolation methods such as:

-Softmax Splatting for Video Frame Interpolation

-Depth-aware video frame interpolation

which also warp context information before image prediction. The authors also didn’t discuss some other important related work such as “Learned Video Compression, ICCV 2019” which also goes a little into the direction of a conditioning not limited to residuals only.

There is also a strong link with [11, 14, 15] but this is sufficiently discussed in the paper

**Quality**

The idea presented in the paper is intuitive and leverages all the knowledge gained in both video compression and frame interpolation. Quantitative evaluation shows state of the art results on a large variety of test sets. I still have the following issues:

-The evaluation only presents a small set of methods. Since it is done on the same datasets it would be important to present other top performing methods for better context. Is there a reason not to include them?

-Only one quality evaluation is presented in supplementary material. Such a figure should appear in the main paper. In addition to this, comparisons with more methods should be included as well.

-Temporal stability is not discussed and the authors didn’t provide a video as supplementary material which I think is quite important to evaluate the work. Also the authors didn’t explain how to maintain similar quality between the key frame and the interpolated frames

-The discussion around the information represented in the feature domain lacks precision. In particular the discussion on the link between the channel 1 in Figure.3 and the motion vectors. Visually one can argue that high intensity values correspond to the “red players”. An ablation study would be required to better understand the content: all values to 0 or set to the average and see the effect on the image.


**Clarity**

In general the paper is clear and easy to follow.

-“MEMC” is used without being introduced. I suppose it stands for Motion Estimation and Motion Compensation? Same comment about “MV”

-l.122: … of network. => “of neural networks”?

-The discussion section is a little rough and would benefit from a revision.

**Significance**

The authors proposed a new approach for learning based video compression which seems to be setting a new state of the art result.

**After Rebuttal**

The authors answered most of my questions and concerns in their rebuttal. I would insist they include more visual and qualitative examples in the revised version.


**Time Spent Reviewing:**

4

---

> ### Author Response · Authors · 2021-08-10
> **Response to the comments from reviewer JwvE**
>
> Many thanks for your valuable comments and the recognition on our contributions. We hope the following responses could address your concerns.
>
> 1. **Reference to prior articles.**
>
>     We will cite these papers and add the related discussion in the revised paper.
>
> 2. **The evaluation only presents a small set of methods. Is there a reason not to include other methods?**
>
>     We have more neural baselines in the supplementary material (Figure 6 in Section 5), including RY_CVPR20, LU_ECCV20, and HU_ECCV20. In the original main paper, we only include the DVC and DVCPro because we could retest DVC and DVCPro using the same intra frame coding method for fair comparison. We will move more comparison results to the main paper.
>
> 3. **Temporal stability is not discussed.**
>
>     We fully agree that the temporal stability is very important for video compression, which is also an advantage of traditional residual coding. We checked the reconstructed videos of the proposed method offline and did not find obvious flickering. Nevertheless, the temporal stability could be further improved by post processing or additional supervision (e.g., loss about temporal stability) at training stage. We will add this discussion in the revised paper.
>
> 4. **How to maintain similar quality between the key frame and the interpolated frames?**
>
>     Error propagation is a common problem in video compression. Frankly speaking, we did not do anything special to maintain the similar quality across frames. Our method has less error propagation because it can achieve better average quality under similar bitrate. As a result, the slope of quality degradation is much lower than residual coding (Figure 5 in the supplementary material) and then the quality of the key frame and the following frames is similar.
>
> 5. **The discussion around the information represented in the feature domain lacks precision.**
>
>     Thanks for your insight. We agree that linking channel 1 with motion vector is one way and with red color is another possible way. Without further study and experiments, it is hard to make a conclusion. We will modify our description in Figure 3 in the revised paper to make it accurate.
>
> 6. **The discussion section is a little rough and would benefit from a revision.**
>
>     We will add more discussions (e.g., about temporal stability) to the main paper in the revised paper.
>
> 7. **Clarity and typos.**
>
>     We will make corresponding modifications in the revised paper.

---

### Official Review · Reviewer_JZR2 · 2021-07-13

**Rating:** 7
**Confidence:** 4

**Summary:**

This paper extends an existing neural video compression method (DVCPro) by adding a two crucial components: (1) the usage of optical flow in warping latents; (2) the usage of optical flow to design a non-additive decoder.  These additions result in a state of the part (as stated by authors, but not verified by me) performance reported on a few (small) datasets usually used for video compression evaluation, outperforming HEVC.

**Main Review:**

This paper tackles the task of neural video coding.

This task, in one of its most common forms takes key-frames, which get encoded/decoded separately, and uses these to seed the start of a sequence of compressed frames. The sequence of compressed P-frames ("predicted") uses the previously decoded frame(s) and motion to come up with a hypothesis for what the current frame should be. The difference (residual) is then encoded in the bitstream.

This paper essentially does exactly this, but with a few crucial twists.

The motion information is used for warping latents (i.e., the quantized tensors which are used for either encoding/decoding or conditioning), which as far as I know is relatively uncommon. These are then used in various places within the encoder/decoder/entropy coder for the P-frame residual.

Secondly, in a typical video compression framework, the output ("decoded") frame is computed as a warp of the previously decoded frame plus the residual ("additive reconstruction"). This typically causes some problems, which this paper avoids by providing the decoder instead with latent values computed from the previously decoded frame, with the decoder's task being to perform a full reconstruction from that and the additional information.

In addition to the hyper-prior (done per frame), the paper also employs a temporal prior model to boost performance.

The authors perform various ablations, which can be found in the supplementary materials. I feel this paper is incomplete by a large margin if we exclude the supplementary, where most of the details about the paper are found.

Overall, I think this is a pretty good paper, but ONLY if we take into account the supplementary. Without them I felt scratching my head multiple times, but all the missing information that made me wonder "why is this not discussed/presented/tested" was present in supplementary.

**Time Spent Reviewing:**

1.5

---

> ### Author Response · Authors · 2021-08-10
> **Response to the comments from reviewer JZR2**
>
> Many thanks for your valuable comments. As you pointed out, the supplementary material includes many details and key information that should be in the main paper. We will revise the paper according to your suggestions.
>
> 1. **This is a pretty good paper, but ONLY if we take into account the supplementary material.**
>
>     After reading your reviews, we also realize that some ablation studies should be put in the main paper. In particular, we will add the following results to the main paper:
>
>     - Comparison with more neural network baselines.
>
>     - Ablation study on spatial prior and temporal prior for entropy model.
>
>     - Ablation study on conditional coding (concatenation) and conditional entropy coder (temporal prior).
>
> 2. **In addition to the hyper-prior, the paper also employs a temporal prior model to boost performance.**
>
>     This is also a key point we missed in the original main paper. We will move it from the supplementary material to the main paper.

---

> > ### Comment · Reviewer_JZR2 · 2021-08-17
> > **reply**
> >
> > Thanks for the reply and be sure to update the paper as suggested.

---

### Official Review · Reviewer_jmWx · 2021-07-15

**Rating:** 5
**Confidence:** 4

**Summary:**

The author propose a video compression framework that leverages context model in place of conventional residual compressor.

**Limitations And Societal Impact:**

yes

**Main Review:**

pros:
1. use a context model to improve the compression performance
2. Better rate-distortion performance over DVCpro baseline

cons:
1. Need more neural baselines.
2. Conditional entropy model is not a novel idea for video compression. (see point 4)
3. Spatial autoregressive prior is not a proper solution for video, as it will significantly increase both encoding and decoding time. Its autoregressive feature makes it hard to be parallelized. In addition, entropy coding usually happens in CPU but network inference happens in NPU/GPU, the accumulated communication cost in autoregressive calls will further worsen the coding efficiency. Given the fact that video decoding is sensitive to runtime (high frame rate), I disagree to use any spatial autoregressive model here.
4. I'm skeptical about the effectiveness of the context model.
    - Author tries to concatenate $\bar{\textbf x}_t$ in en/decoder to demonstrate that it's better than residual, but I didn't see there is any theoretical difference. This can simply be regarded as a different way to incorporate residual, and the concatenation method requires more model parameters and conv layers. The invertibility of the original residual also makes the model easier to train.
    - Similar idea was already proposed to improve the compression (Yang et al.). They interpret the residual compression as an invertible transform (or normalizing flow) and propose a similar(also simpler) conditional prior.
    - $\textbf{Comments}$: The context $\bar x$ also acts as a condition of the conditional entropy coder. As the concatenation method is simply a different form of residual, I think the performance improvement may mainly come from the conditional entropy coder. Extra conditioning makes the prior fit the distribution of the latent variable better (for better probabilistic model); residual is a cheap way to reduce the redundancy and removing high-frequency information (for better transform). They are aimed at solving the same problem in slightly different ways.

Yang, Ruihan, et al. "Hierarchical Autoregressive Modeling for Neural Video Compression." International Conference on Learning Representations. 2021.

**Time Spent Reviewing:**

2

---

> ### Author Response · Authors · 2021-08-10
> **Response to the comments from reviewer jmWx**
>
> Many thanks for your valuable comments. We gave more detailed comparisons in the supplementary material, which should be highlighted in the main paper. We hope the following responses could address your concerns.
>
> 1. **Compare with more neural baselines.**
>
>     We gave more neural baselines in the supplementary material (Figure 6), including RY_CVPR20, LU_ECCV20, and HU_ECCV20. Thank you for pointing it out. We will move these results from the supplementary material to the main paper.
>
> 2. **Spatial autoregressive prior is not a proper solution for video.**
>
>     We agree that the non-parallelable autoregressive prior cannot be easily used in practical applications. It is also the main reason that we propose using temporal prior in our entropy model. In the supplementary material (Figure 8), we gave the ablation study. Below please find the BD-rate comparison:
>
>     |Entropy model| Bitrate increase|
>     |-|-|
>     |hyper prior + spatial prior + temporal prior|0.0%|
>     |hyper prior + spatial prior|4.6%|
>     |hyper prior + temporal prior|3.8%|
>     |hyper prior|60.9%|
>     | | |
>
>     From this table, we can find that removing spatial autoregressive prior but keeping the temporal prior leads to small loss (3.8%).
>     Thanks for pointing it out. We will revise the main paper accordingly.
>
> 3. **Difference between concatenation and fixed subtraction/add.**
>
>     It is important to explore both spatial and temporal correlation in a modern video codec. The traditional codec employs intra and inter prediction modes at block level to adaptively explore the spatial and temporal correlation via rate distortion optimization. However, most existing DL-based codecs rely on fixed subtraction/add operations at frame level. It is suboptimal for new content regions because the energy of residual via subtraction may be larger than that in the original pixels. By contrast, our conditional coding tends to adaptively explore the spatial correlation for these regions. In Figure 3, we show that the conditional coding using concatenation can significantly reduce the reconstruction error at the new content regions when compared with residual coding.
>
> 4. **Difference with Yang et al. 2021.**
>
>     Thank you for pointing us to this related paper. We think Yang et al. 2021 and our paper share some similar insight but solve the problem from different angles.
>     The main differences between our paper and Yang et al. 2021 include:
>
>     - In Yang et al. 2021, the encoder input is residual, and the residual is adaptively scaled. In our method, we do not rely on residual compression. The encoder input is the current frame concatenated with the context feature.
>     - In Yang et al. 2021, the extra conditional prior is from flow latents. In our entropy model, the extra conditional prior is from the context feature extracted from the previous frame.
>
>     We will cite this paper and add the related discussion.
>
> 5. **Performance improvement may mainly come from the conditional entropy coder.**
>
>     We agree that it is important to show the effectiveness of conditional coding using concatenation. From below experimental results, we observe that conditional coding (concatenation only) has demonstrated its superior performance with coding gain even larger than that of conditional entropy coder (temporal prior only). We will add these results in the revised paper.
>
>     |Temporal prior|Concatenate context feature|Bitrate increase|
>     |:-:|:-:|:-:|
>     |Y|Y|0.0%|
>     | |Y|4.6%|
>     |Y||11.2%|
>     | | | 12.9%|
>     | | | |

---

### Decision · Program_Chairs · 2021-09-27

**Decision:**

Accept (Poster)

**Comment:**

The authors propose a new video compression model that uses motion vectors to extract and warp features instead of warping the previous frame for predicting the current frame. As a result, the encoder doesn’t rely on residual compression but directly encodes the necessary information needed to decode the frame.

The paper leads to a new state of the art in video compression and merits publication. Unfortunately, the main paper lacked several citations and baseline comparisons. Some of these comparisons were shown only in the appendix, making the main paper appear incomplete. I strongly urge the authors to present these results in the main paper. In addition, I also encourage the authors to run additional baseline comparisons of the related works mentioned by the reviewers (such as Ruihan Yang et al., ICLR 2021).